# Energy-Efficient Wireless Communication Strategy for Precision Agriculture Irrigation Control

**DOI:** 10.3390/s21165541

**Published:** 2021-08-18

**Authors:** Camilo Lozoya, Antonio Favela-Contreras, Alberto Aguilar-Gonzalez, L.C. Félix-Herrán, Luis Orona

**Affiliations:** 1Tecnologico de Monterrey, School of Engineering and Science, Ave. Heroico Colegio Militar 4700, Chihuahua 31300, Mexico; alberto.aguilar@tec.mx (A.A.-G.); l.orona@tec.mx (L.O.); 2Tecnologico de Monterrey, School of Engineering and Science, Ave. Eugenio Garza Sada 2501, Monterrey 64849, Mexico; antonio.favela@tec.mx; 3Tecnologico de Monterrey, School of Engineering and Science, Blvd. Enrique Mazón López 965, Hermosillo 83000, Mexico; lcfelix@tec.mx

**Keywords:** energy-efficient communication, wireless sensor network, self-triggered control, precision agriculture

## Abstract

In smart farming, precision agriculture irrigation is essential to reduce water consumption and produce higher crop yields. Closed-loop irrigation based on soil moisture measurements has demonstrated the capability to achieve a considerable amount of water savings while growing healthy crops. Automated irrigation systems are typically implemented over wireless sensor networks, where the sensing devices are battery-powered, and thus they have to manage energy constraints by implementing efficient communication schemas. Self-triggered control is an aperiodic sampling strategy capable of reducing the number of networked messages compared to traditional periodical sampling. In this paper, we propose an energy-efficient communication strategy for closed-loop control irrigation, implemented over a wireless sensor network, where event-driven soil moisture measurements are conducted by the sensing devices only when needed. Thereby, the self-triggered algorithm estimates the occurrence of the next sampling period based on the process dynamics. The proposed strategy was evaluated in a pecan crop field and compared with periodical sampling implementations. The experimental results show that the proposed adaptive sampling rate technique decreased the number of communication messages more than 85% and reduced power consumption up to 20%, while still accomplishing the system control objectives in terms of the irrigation efficiency and water consumption.

## 1. Introduction

Smart farming aims to reduce costs and increase productivity while optimizing natural resources through the processing and analyzing of data obtained from sensor devices [1]. Hence, its implementation requires higher demands for data collection and network communication [2]. Perception, interconnection, and decision-making, through the use of modern technology, comprise the three main tasks for precision agriculture [3]. Crop and environmental variables, such as soil moisture, leaf wetness, air temperature, atmospheric pressure, relative humidity, and solar radiation, are commonly monitored in different agriculture applications [4].

Soil moisture may be the most complex variable to measure due to its high spatial and temporal variability and nonlinear behavior. The relevance of measuring the soil moisture has recently increased as closed-loop irrigation, based on this variable, has become one of the most effective strategies to improve water efficiency by applying the right amount of water to crops at the right time [5].

Typically, closed-loop irrigation systems collect data via the implementation of a wireless sensor network (WSN) over large crop area fields with multiple sensing devices placed in different locations to obtain representative soil moisture values. During the last ten years, different works [6,7,8,9,10,11,12] demonstrated the relevance of soil moisture measurements implemented over a WSN to achieve effective and optimum water utilization for irrigation in large crop areas.

Implementing WSNs for agricultural applications requires efficient energy management since several battery-powered sensing devices may comprise the network [13]. Energy is mainly dissipated during communication; therefore, power reduction can be achieved by implementing a sleep/wake-up strategy [14], where wireless devices enter standby mode and stop communication until the next transmission time.

One of the early used techniques is duty cycle scheduling [15], where nodes are programmed to turn on and turn off on a pre-established basis; however, this approach presents a trade-off between energy efficiency and sensing quality. Recently, adaptive sampling rate techniques based on process dynamics have accomplished significant energy-efficient improvements by activating transmissions only when relevant measurements are required [16].

From a control perspective, digital closed loops have been traditionally implemented using periodical sampling. However, event-driven aperiodic strategies began gaining attention due to the efficiency improvements achieved in the processing time and data communication [17]. Furthermore, event-driven control demonstrated a considerable reduction in communication messages while still achieving the control objectives [18]. Specifically, self-triggered control (STC) implements model-based algorithms to estimate the next sampling event by predicting the instant where the following measurement should take place based on process dynamics [19]. Recently, different self-triggered implementations over WSN have been proposed [18,20,21,22,23], obtaining relevant control performance and communication efficiency results. In every case, the experiments were either simulated or conducted on laboratory conditions where the controlled processes had well-known state–space models.

The work presented in this paper proposes an adaptive sampling rate energy-efficient communication strategy to reduce power consumption from the remote nodes in a wireless sensor network using self-triggered algorithms to implement closed-loop control for precision agriculture irrigation. The objective is to reduce the number of messages in the network by transmitting sensor data only when relevant while accomplishing the irrigation control objectives of using water efficiently and avoiding unhealthy crops due to water stress. The proposed strategy was evaluated and compared with duty cycle periodical-scheduling implementations in four irrigation areas of a pecan crop experimental field.

Different energy management schemes for WSN in agricultural and environmental monitoring applications have been proposed by the research community, as summarized by [14,24]. From a pure communication perspective, techniques, such as efficient routing protocols, efficient medium access control, and data-clustering, have been widely used to reduce power consumption, as demonstrated by [25,26,27,28,29,30,31]. On the other hand, the sleep/wake-up method adopted by the duty cycle scheduling and the adaptive sampling rate techniques consider a control-communication approach.

In this regard, the work in [32] proposed a power reduction algorithm based on redundant data for an agriculture irrigation system where data transmission is activated after the difference between the soil moisture readings reaches a specific threshold. In this case, moisture dynamics determine the adaptive sampling rate; however, the defined threshold has a fixed value that does not consider the crop-weather characteristics. In [33], the authors analyzed different duty-cycling mechanisms, including the scheduled and the on-demand approaches. In the first one, the node periodically wakes up and communicates; in the latter, the node is awakened as necessary by an external event. Although the on-demand technique can achieve better results, it relies on an additional communication channel.

In [34], the authors proposed an improved duty cycling algorithm to reduce the energy consumption in an agriculture field. The algorithm determines the on–off periods based on energy data aggregation at the base station; however, it does not consider the agriculture process dynamics variables. The work in [35] implemented a content awareness communication system where a sufficiently knowledgeable controller manages a WSN by determining whether each sensor should transmit data or not; however, as stated by the authors, this centralized approach may increase the network load.

In [36], an adaptive duty cycle scheme was proposed based on the queue size and priority class of a packet to reduce the delay of high priority packets and support time-bounded delivery of priority packets. This approach concentrates on energy savings based on purely communication parameters.

The model proposed by [37] describes soil, crop, and weather dynamics for a grass irrigation control system, and we enhanced and applied it to a pecan crop field to implement an adaptive sampling rate sleep/wake-up strategy based on self-triggered control to implement an energy-efficient algorithm to communicate the sensor nodes over a WSN.

The main contributions of this paper are: (1) the identification and validation of the specific process dynamics for the pecan crop irrigation system, (2) the implementation of a model-based self-triggered control algorithm to decrease the number of communication messages while still accomplishing control objectives, and (3) the estimation of the energy consumption improvement in the wireless sensor nodes due to the use of this control approach in comparison with the duty-cycle periodical scheduling technique.

The rest of this paper is organized as follows: In Section 2, the elements that comprise the closed-loop irrigation system are described, and the irrigation state–space model is identified and validated with measured data. In Section 3, the implementation of the proposed energy-efficient communication strategy based on self-triggered control is detailed. In Section 4, the experimental control performance and power consumption results are presented and discussed. Section 5 summarizes the conclusions and describes future work.

## 2. Precision Irrigation Control System

An automated precision irrigation process can be implemented using soil moisture readings in the crop-root zone to feedback the system; meanwhile, weather conditions represent the system disturbance measured to compensate their effects in advance [9]. The soil moisture indicates the percentage water content available in a specific soil volume sample and is often referred to as the volumetric water content (VWC). The reference evapotranspiration (ETO) represents the water loss caused by weather conditions for a specific reference crop (grass). A crop coefficient $K_{c}$ is required to convert this reference value into a crop specific parameter. Solar radiation, wind speed, air temperature, and relative air humidity measurements are required to calculate this parameter, according to the FAO Penman–Monteith equation [38]. The evapotranspiration is generally expressed in millimeters (mm) of water lost from a crop surfaced per unit of time (days).

### 2.1. State Space Model

The hydrological balance model describes the dynamics of an irrigation system where the soil moisture changes in the root zone $\dot{\theta }$ were obtained from the sum of water inflows and outflows. If we consider a plain field (no water runoff) in a dry area (no rainfall and no water capillary rise), as stated by [37], then the system behavior is expressed as
(1)$$\dot{\theta }\left(t\right)=a_{1}ir\left(t-\tau \right)-K_{c}eto\left(t\right)-a_{2}\theta \left(t\right),$$
where ir is the applied irrigation, eto is the reference evapotranspiration, and $a_{1}$, $K_{c}$, and $a_{2}$ are constants that, respectively, denote the irrigation efficiency, evapotranspiration coefficient to incorporate crop characteristics, and soil moisture proportionality to estimate deep percolation. In addition, τ is the time delay required by the water to reach the crop roots, which usually requires several minutes.

Since a discrete model is required, Euler approximation is used to discretize Equation (Equation 1) with a sampling period *h*, and then
(2)$$\theta \left(kh+h\right)=c_{1}\theta \left(kh\right)+c_{2}ir\left(kh-\tau \right)+c_{3}eto\left(kh\right),$$
where $c_{1}$, $c_{2}$, and $c_{3}$ are discrete coefficients that absorb the previous continuous coefficients and signs.

The model is enhanced by considering soil moisture variations as a second state variable; therefore, δ, defined as
(3)δ(kh+h)=θ(kh)−θ(kh−h),
represents the relative moisture dynamics and indicates the changes detected in the root zone. Now, the state–space model for an irrigation system is specified as a second order system in the general form of
(4)$$\begin{array}{cc} x(kh+h) & =\Phi x(kh)+\Gamma u(kh)\\ y(kh) & =Cx(kh), \end{array}$$
where the state and input and output vectors are defined, respectively, as
(5)$$x\left(kh\right)=\left[\begin{array}{c} \theta (kh)\\ \delta (kh) \end{array}\right],u\left(kh\right)=\left[\begin{array}{c} ir(kh-\tau )\\ eto(kh) \end{array}\right],y\left(kh\right)=\left[\begin{array}{c} \theta (kh) \end{array}\right],$$
while the discrete matrices $\Phi \in R^{2\times 2}$, $\Gamma \in R^{2\times 2}$, and $C\in R^{1\times 2}$ define the process dynamics coefficients to be identified.

### 2.2. Networked Control System

Efficient irrigation management requires real-time soil moisture measurements to capture temporal and spatial variations in the crop root zone. However, obtaining representative and accurate readings over large crop areas is a challenging task since the soil hydraulic properties and spatial heterogeneity heavily influence the moisture and crop rooting patterns, among other variables [39]. Moisture sensors use electromagnetic principles to measure the water content; furthermore, to obtain accurate values, soil-specific calibration is required before installation.

The sensor 10HS (METER Group Inc., Pullman, WA, USA) is a low-cost VWC sensor with a measurement volume range of 1
${\mathrm{dm}}^{3}$, and according to [40] measurement error can be reduced from 5% to 1% if off-line calibration is conducted previous to installation. The experimental field consists of one hectare of pecan crop (*Carya illinoinensis*) divided into four irrigation areas of 25 m × 100 m, as illustrated in Figure 1. Each area contains approximately 45 productive trees watered with a micro-sprinkler irrigation system. The field is located in northern Mexico (Chihuahua city), which is an arid region with very low annual rainfall.

As shown in Figure 2, the networked control system comprised one sensor node and one actuator node for each irrigation area; there was one weather node and a single controller for the entire experimental field. The WSN was implemented over the IEEE 802.15.4 standard, which is the basis for the Zigbee communication protocol [41]. Zigbee has become a popular protocol due to its low cost, low power consumption, and small communication packet size. Using a client-server communication schema, each sensor node sends soil moisture data to the controller node, the controller uses the weather and soil moisture information to determine if irrigation is needed, and then the control message is sent to the corresponding actuator node.

Each sensing node includes three 10HS sensors buried in the shallow level of the pecan effective root zone (60 cms), with a horizontal separation of 3 m among them; the three readings’ average value provides a representative soil moisture measurement for the irrigation area. The sensors were previously calibrated for the specific soil texture (silty clay loam) according to the method described by [42]. An irrigation electrovalve activates and deactivates field watering for each actuator node, and a flow sensor measures the water consumption. The weather node includes a solar radiation sensor (PYR sensor from METER Group Inc., USA), a wind velocity sensor (Davis Cup sensor from METER Group Inc., USA), and a temperature–humidity sensor (VP4 sensor from METER Group Inc., USA) to calculate the reference evapotranspiration for the crop field.

These three types of nodes were built upon the Arduino MKR board (Arduino, USA), which includes a 32-bit microprocessor and a 12-bit analog-to-digital converter. The board was energized with a 3.3-V power supply, and it was capable of reducing the current consumption from 20 mA in operation mode to 0.57 mA in standby mode. The controller node was implemented with a Raspberry Pi 3 Model B (Raspberry Pi Foundation, Cambridge, UK) based on an ARM Cortex-A53 64-bit processor. The controller ran four simultaneous control tasks to implement the four loops of the irrigation system.

### 2.3. Process Identification and Model Validation

Soil moisture dynamics have complex non-linear behavior; however, a linear model can be approximated by identifying the main soil moisture regions delimited by the soil field capacity (FC) and crop permanent wilting point (PWP) thresholds. The FC refers to the level where the soil can no longer retain water, and the PWP is the limit where below this, the crop cannot absorb water from the crop. As shown in Figure 3, the FC and PWP define three soil moisture regions with different dynamics: gravitational water, available water, and unavailable water.

To properly model the soil moisture dynamics, different values for the matrices Φ and Γ must be identified appropriately in model Equation (Equation 4) for each region. The maximum allowable depletion (MAD) is the reference value for crop irrigation typically located in the middle of the available water region; below this (negative soil moisture values), the crop will become water-stressed, while, above this (positive soil moisture values ), water may be wasted. Sensor node readings have been normalized to provide soil moisture measurements as a relative percentage value to this reference point.

Over four weeks, irrigation, soil moisture, and reference evapotranspiration data were gathered at a sampling rate of 10 min, since, as suggested by [4], this represents, in practice, the minimum sampling time to capture the irrigation process dynamics. The obtained data was divided according to the soil moisture regions, and then coefficients values for Φ and Γ were identified using the estimation algorithm proposed by [9] to model the process dynamics. Table 1 shows the obtained coefficients for each irrigation area.

A new four-week dataset was acquired from the networked control system to validate the identified model. We compared the latest measured data with the estimated soil moisture values obtained from the model by using the correlation coefficient $R^{2}$ defined as
(6)$$R^{2}=1-\;\frac{\sum_{i=1}^{n}{\left(\theta_{i}-{\hat{\theta }}_{i}\right)}^{2}}{\sum_{i=1}^{n}{\left(\theta_{i}-\bar{\theta }\right)}^{2}},$$
where $\theta_{i}$ represents the soil moisture readings, $\hat{\theta_{i}}$ represents the model-based estimated values, $\bar{\theta }$ is the mean value from all measurements, and *n*, with a value of 4032, is the number of measurements for the validation period.

Figure 4 shows the correlation between the estimated soil moisture values calculated from the state–space model ($\hat{\theta }$) and the measured soil moisture values obtained from the sensor readings (θ). There was no automatic irrigation control during the validation process, and we applied irregular watering to cover the different soil moisture regions. Thus, during week 2, intensive irrigation shifted the moisture values to the gravitational region, and, at the end of week 3, the lack of irrigation shifted the values to the unavailable region. The model output was not fed with measured data to avoid any correction during the four weeks. The correlation results ($R^{2}=0.8616$) showed that the identified model adequately captured the irrigation dynamics for the pecan crop.

## 3. Energy-Efficient Communication

Once the pecan crop irrigation model was appropriately identified and validated, we implemented the WSN communication strategy. Energy-efficient communication was achieved by reducing the number of communication messages by the sensor node. This node included a triggered mechanism to estimate the next sampling instance h+1 based on soil moisture dynamics θ and weather conditions eto. The control system implemented a feedback/feedforward loop where soil moisture was the process variable, while weather conditions disturbed the process dynamics. The control node sent an on–off signal ir to the actuator node to manipulate the irrigation valve, based on the difference between the reference level *r* and the measured variable θ, as depicted in Figure 5. The irrigation was activated when the error was below −2% and deactivated when it was above +2%.

### 3.1. Self-Triggered Control

Closed-loop control algorithms have been traditionally implemented using periodical sampling. Although periodicity simplifies the control system design and analysis, it also leads to the unnecessary use of communication, energy, and computation resources in the system devices. Aperiodic sampling avoids this situation by conducting measurements only when a relevant event is about to happen, and it can be implemented using either a reactive (event-based) or a proactive approach (self-triggered). In the first one, the sampling instant is determined when a monitored output variable reaches a specific threshold, while, in the latter one, the next update time is predicted based on the evolution of the process state variables.

Self-triggered control (STC) has an advantage over event-triggered control (ETC) since it does not require dedicated additional energy-consuming hardware to conduct continuous measurements. Instead, it uses process dynamics knowledge to implement the model-based algorithm to estimate the next sampling event. We considered two different self-triggered strategies: the first one calculates the measured error to estimate the next sampling instant [17], while the second one uses the Lyapunov function to trigger the control updates [20].

In the error-based triggering condition $\gamma_{1}$, the process states were forced to stay within a given error defined as a function of the norm of the measured states; thereby, the next sampling time h+1 was obtained when the measured error exceeded the limit that was proportional to the current process state. Thus,
(7)$$\gamma_{1}:\;h+1\;=\;min\;\left\{\;h\leq max\_limit\;\;\;|\;\;\;\left|e\left(h\right)\right|>\sigma \;\left|x\left(kh\right)\right|\;\;\right\},$$
where 0<σ≤1 denotes proportionality, max_limit is the maximum allowable sampling period, and the measured error is defined as
(8)$$e\left(h\right)=\;x\left(kh+h\right)-x\left(kh\right),$$
which represents the difference between the estimated next state value with respect to the current one.

In the Lyapunov-based triggering condition $\gamma_{2}$, the estimation of the next sampling instant is determined by evaluating the Lyapunov decaying rate over time ΔV. Given a linear discrete control model defined by Equation (Equation 4), the system is asymptotically stable in the Lyapunov sense if there is a function *V* that satisfies
(9)$$V\left(x(kh)\right)=\;{x(kh)}^{T}\;P\;x\left(kh\right)>0,$$
and
(10)$$\Delta V\left(x(kh)\right)=\;{x(kh)}^{T}\;\left(\Phi^{T}P\Phi -P\right)\;x\left(kh\right)<0,$$
where
(11)$$\Delta V\left(x(kh)\right)=\;V\left(x(kh+h)\right)-V\left(x(kh)\right),$$
the discrete-time Lyapunov matrix *P* must be a positive definite matrix, and its existence certifies the system stability since *V* is always decreasing. To ensure that the Lyapunov function rate is proportional to the current value, then the next sampling time h+1 is obtained when the estimated Lyapunov function decays under a limit proportional to the current function value. Thus,
(12)$$\gamma_{2}:\;h+1\;=\;min\;\left\{\;h\leq max\_limit\;\;\;|\;\;\;V\left(x\left(h\right)\right)<\sigma V\left(x\left(kh\right)\right)\;\;\right\},$$
where 0<σ≤1 also denotes proportionality.

In any case, if the plant dynamics for the crop irrigation process are known, then the evolution of the state variables can be estimated to predict the next update time from the last measurement.

### 3.2. Algorithm Design

The proposed energy-efficient algorithm ran in each sensor node to communicate soil moisture readings using the sleep/wake-up strategy. The algorithm implemented the triggering conditions defined by Equations (Equation 7) and (Equation 12) according to the specific dynamics of the pecan crop irrigation system. First, we describe how self-triggered constant values for σ and *P* were selected, and then we explain how the sensor node executed the algorithm to estimate the next sampling instance based on the soil moisture dynamics specified by Equations (Equation 4) and (Equation 5), with the identified coefficients denoted in Table 1.

To select suitable values for the σ constant, we conducted four-week off-line simulations with different initial conditions. As seen in Figure 6 there was a trade-off between the number of communication messages and the estimated water consumption by the irrigation system. Therefore, we selected the σ values of 0.10 and 0.60 for triggering conditions $\gamma_{1}$ and $\gamma_{2}$ respectively, where the message’s curve was in the inflection point, and the expected water consumption was in the lower values.

From Equations (Equation 9) and (Equation 10), we observed that, for *V* to be a Lyapunov function, it is necessary and sufficient that there exist a positive definite matrix *P* that satisfies
(13)$$\Phi^{T}\;P\;\Phi -P<-Q,$$
where *Q* is positive definite. One way of determining the Lyapunov function is to choose a positive definite matrix *Q* and then obtain *P* from Equation (Equation 13), if *P* is positive definite then the system is asymptotically stable [43]. Typically an identity matrix is chosen for *Q*; however, for this specific problem, we assigned more weight to state variable δ than variable θ, since we were more interested in increment sampling based on soil moisture variations rather than the soil moisture value. Therefore, to calculate *P* for any value of Φ given by Table 1, we use matrix
(14)$$Q=\left[\begin{array}{cc} 0.002 & 0\\ 0 & 1.5 \end{array}\right].$$

For the algorithm, each node is required to wake up, take soil moisture readings, transmit data to the controller, estimate the next sampling instant, and go to sleep again, as described by Algorithm 1. Function read_input_sensors() conducts the measurement from the three sensors providing an average soil moisture value, then select_ coefficients() obtain from a lookup table the proper values for matrices Φ and Γ according to the moisture region. Finally, the sensor node communicates with the controller and calculates the time for the next sampling instance before going into standby mode.

For periodic sampling, the function that calculates the next sampling instant returns a fixed time value. For aperiodic sampling, a function in Algorithm 2 implements the error-based triggering condition $\gamma_{1}$ and Algorithm 3 describes the Lyapunov-based condition $\gamma_{2}$. In the Lyapunov approach, function select_P_matrix() obtains from a lookup table the pre-calculated value of *P* that solves Lyapunov Equation (Equation 13). In self-triggered control, usually, there is an upper bound sampling period in order to provide robustness guarantees for the controller, for the current implementation max_limit is defined with a value of 120 min in order to ensure that the sampling period does not go beyond this limit. This value limits the number of cycles in the loop, thus, avoiding excessive processing loads.
**Algorithm 1:** Sensor node algorithm.
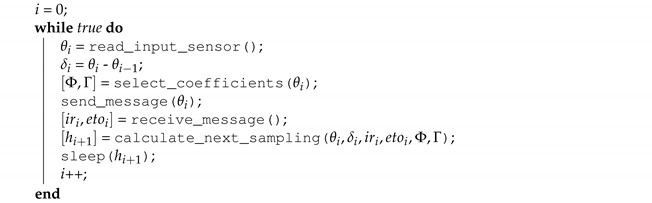


**Algorithm 2:** Next sampling estimation based on error measurements.

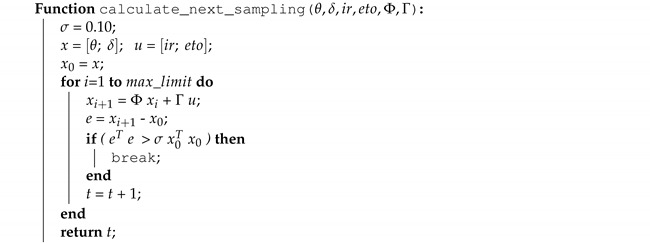



**Algorithm 3:** The next sampling estimation based on Lyapunov dynamics.

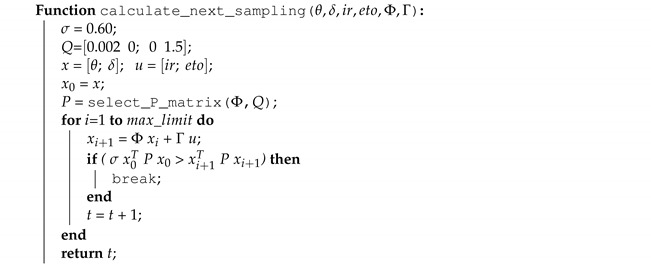



## 4. Results and Discussion

This section presents the experimental results obtained from the performance of the closed-loop irrigation system during an evaluation period of four weeks. First, we show the results of each one of the four sampling strategies according to the evaluation parameters. Then, we analyze each transient response to identify the differences for every sampling algorithm. Finally, based on the experimental results, we estimate the power consumption for each strategy. It is important to highlight that, as a result of the use of four different sampling strategies; there were no significant differences in the vegetative evolution of the crops; this means that, in general, the control system worked correctly in the four irrigation areas.

### 4.1. Evaluation Parameters

Two periodical (duty cycle scheduling) and two aperiodical (adaptive sample rate) sampling strategies were considered for the experimental evaluation:$\gamma_{0}$, a time-triggered approach with a sampling period of 10 min.$\gamma_{1}$, a self-triggered approach based on error measurement.$\gamma_{2}$, a self-triggered approach based on Lyapunov dynamics.$\gamma_{3}$, a time-triggered approach with a sampling period of 80 min.

The periodical sampling of 10 min represents the minimum practical time interval typically used in agriculture, while the 80 min periodical strategy has an average of a similar number of sampling events compared with the self-triggered approaches. The experimental evaluation was conducted simultaneously in the four irrigation areas, depicted in Figure 1, for four weeks. The evaluated strategies were compared in terms of:$J_{1}$, the number of communication messages initiated by the sensor node during the evaluation period.$J_{2}$, the accumulated water consumption during evaluation in m ${}^{3}$.$J_{3}$, the mean squared error or irrigation error obtained from
(15)$$J_{3}=\frac{1}{n}\;\sum_{i=1}^{n}{\left(\theta_{i}-MAD\right)}^{2},$$
where $\theta_{i}$ is the current soil moisture measurement, and the maximum allowable depletion MAD is the soil moisture reference value for the irrigation process. This parameter represents how precise the irrigation is conducted, lower values means higher precision.

### 4.2. Experimental Results

Table 2 shows the experimental results in terms of the number of messages, water consumption, and irrigation error.

Both self-triggered approaches $\gamma_{1}$ and $\gamma_{2}$ significantly reduced the number of communication messages compared with the periodical strategy $\gamma_{0}$, and still achieved practically the same control performance in terms of water consumption and irrigation error. Periodical strategy $\gamma_{3}$ had a similar number of communication events compared with the self-triggered strategies; however, the water consumption was considerably larger, and it also had a higher irrigation error.

Notice that self-triggered algorithms effectively reduced the communication messages while achieving the control objectives. When comparing both self-triggered implementations, the Lyapunov-based algorithm better reduced the water consumption, while the error-based algorithm obtained slightly better results in terms of communication messages and irrigation error. However, these differences are marginal, and the minimum adjustment in the configuration parameters (σ, *P*) would impact these results.

### 4.3. Transient Response

Transient response analysis shows the correlation between the sampling period *h* and the evolution of the process state variables $x_{1}$ (soil moisture) and $x_{2}$ (soil moisture variations) during irrigation events. Figure 7, Figure 8 and Figure 9 depict four-day transient responses for the sampling strategies $\gamma_{1}$, $\gamma_{2}$, and $\gamma_{3}$ respectively.

In both self-triggered strategies (Figure 7 and Figure 8), the sampling instants became more frequent either when the soil moisture $x_{1}$ reached the set point (zero value) or when the soil moisture variation $x_{2}$ registered larger changes due to irrigation. Otherwise the sampling period reached the maximum limit, which was set to 120 min. For the time-triggered approach (Figure 9), the sampling period was always fixed regardless the value of the state variables.

### 4.4. Power Consumption

To estimate the power consumption in the sensor nodes, we conducted current measurements during the different operating states of the node; we also computed the time spent in each state during the sampling events. Table 3 shows the obtained measurements’ average values for each one of the five identified states.

**Data acquisition**. Refers to the actions required for the three sensor reads, including a pulse to energize each sensor, which should be stabilized for at least 100 ms before the measurement and 50 ms after the reading.

**Data transmission**. The microcontroller sends data frames to the RF transceiver containing address and payload information. Then, the transceiver validates the frames, extracts and complements the transmitting information, and finally transmits the data and waits for each packet’s success/failure status. Transmission retries are automatically executed as required.

**Data receiving**. Upon data reception, the RF transceiver identifies the source address of each received packet, sends an acknowledgment message, and then composes the receiving data frame to send it to the microcontroller for further processing.

**Data processing**. Includes all the processing and calculation activities executed by the microcontroller, such as obtaining the average soil moisture value, creating data frames for transmissions, extracting values from receiving data frames, executing the next sampling instance estimation algorithm, and preparing and recovering from the standby mode.

**Standby mode**. Refers to the energy-saving mode for the microcontroller and the RF transceiver.

Based on the previous measurements and considering the number of communication events registered in each sensor node during the four-week experimental evaluation period, the estimated average power consumption in Watts is shown in Figure 10. In addition, we also present the irrigation water consumption for each case, since it is a critical performance parameter and is heavily influenced by the sampling strategy.

The flexibility provided by the aperiodical approaches $\gamma_{1}$ and $\gamma_{2}$ that sampled only when necessary reduced around 20% in power consumption and irrigation water compared to the 10-min and 80-min periodical implementations, respectively.

### 4.5. Discussion

Self-triggered is a model-based control approach, and thus its implementation requires estimating the evolution of the process state variables to predict the next relevant event to communicate data. This capability allows the self-triggered strategies to perform better than the periodical implementations due to adjusting the sampling period according to the irrigation process conditions. Moreover, if the sampling period is set to save water, the device power consumption increases due to a significant amount of messages.

On the other hand, if the periodical strategy is tuned to reduce the frequency of communications messages, this negatively affects the control performance regarding the water consumption and, to a lesser degree, the irrigation error. Self-triggered algorithms provide a balanced solution where the control objectives are fully accomplished and energy efficiency is achieved due to communication messages being transmitted only when needed.

No significant performance difference between both self-triggered algorithms was detected. Therefore, it is difficult to identify which one is more suitable for agriculture applications and what advantages one may provide over the other. The tuning parameters of each algorithm were selected empirically through simulations; however, an analytical solution should be explored. Thus, a more profound analysis must be conducted on self-triggered strategies optimizations for precision agriculture applications.

Another critical aspect that makes viable the implementation of self-triggered algorithms in precision agriculture applications is that the inherent low dynamics of agriculture allows real-time execution regardless of the slight increase of the computation time required to calculate the next sampling instant. This additional processor load is considerable lower—practically insignificant—when compared with the time spent in the communication events. Smart farming relies on the use of WSN to communicate data from sensor devices. The results obtained in this work encourage the implementation of self-triggered algorithms to provide an energy-efficient wireless communications strategy to precision agriculture and other environmental monitoring applications where a state–space model can appropriately describe the process dynamics.

## 5. Conclusions and Future Works

We described and evaluated the design and implementation of self-triggered control algorithms to communicate soil moisture measurements in a closed-loop agriculture irrigation process against the traditional fixed-periodical sampling approach. First, a specific model for a pecan crop irrigation system was identified and validated. Then, four sampling strategies were evaluated simultaneously in the experimental pecan field. Two self-triggered aperiodical sampling algorithms were implemented, considering the state variables to determine the next sampling instant.

One algorithm selected the sampling period based on the estimated measured error, while the other used a Lyapunov function to trigger the communication event. The other two implemented strategies used periodical sampling with low and high fixed communication rates. The experimental results showed that the self-triggered approaches provided higher efficiency in communication messages, thus, implying a significant reduction of energy consumption—a critical parameter for battery-powered wireless nodes. Both self-triggered strategies achieved the control performance objectives regarding water consumption and precision irrigation. This approach demonstrates that the sampling strategy to measure and communicate only when relevant provides an efficient communication mechanism in terms of energy while still achieving the control objectives.

In future works, the implementation of self-triggered techniques in wireless sensor networks can be extended to other agriculture and environmental applications where specific model-based processes can be properly identified to achieve an energy-efficient communication strategy. The triggering conditions for the algorithms can be refined to guarantee optimal performance under specific evaluation parameters. Finally, self-triggered control strategies can be compared with other energy-efficient sampling rate adaptation techniques within the precision agriculture domain.

## Figures and Tables

**Figure 1 sensors-21-05541-f001:**
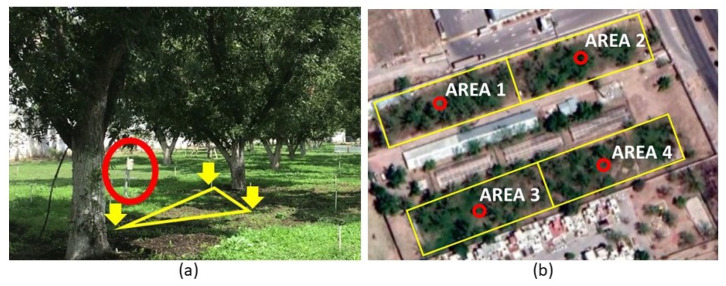
Experimental pecan crop field with the sensor node locations illustrated with the red circles: (**a**) Area 1 lateral view; yellow arrows indicate the sensor placements. (**b**) Satellite view.

**Figure 2 sensors-21-05541-f002:**
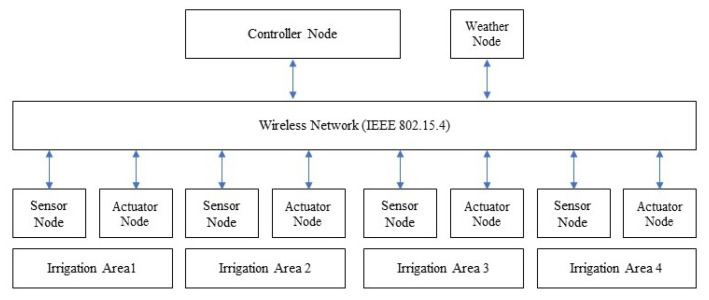
Networked control system for precision agriculture irrigation.

**Figure 3 sensors-21-05541-f003:**
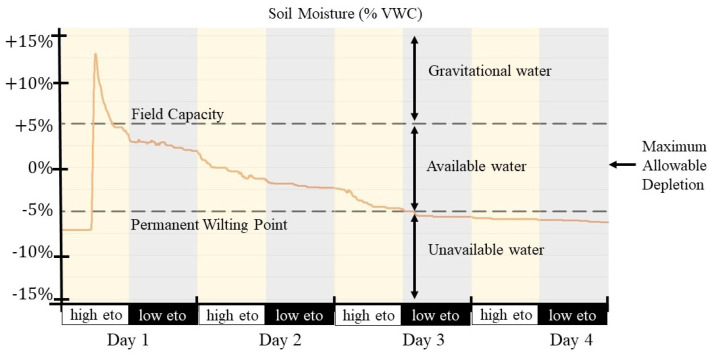
Soil moisture dynamics for the different regions, figure obtained from [9].

**Figure 4 sensors-21-05541-f004:**
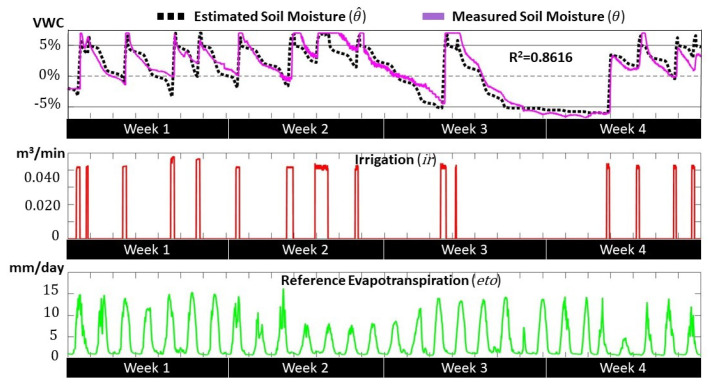
Validation results for process identification.

**Figure 5 sensors-21-05541-f005:**
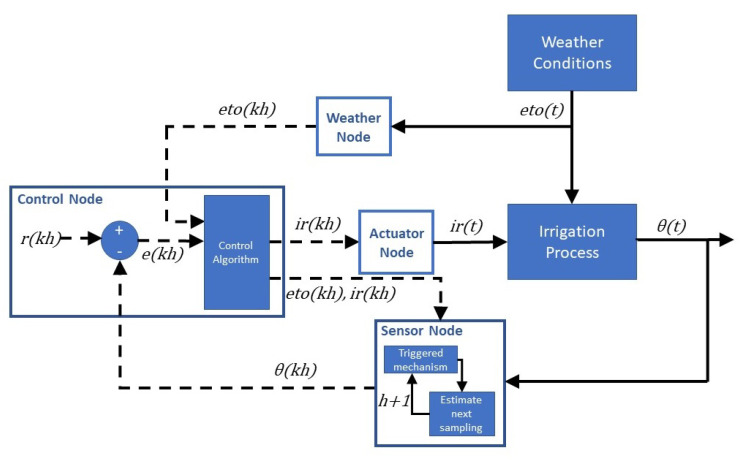
Precision irrigation control loop.

**Figure 6 sensors-21-05541-f006:**
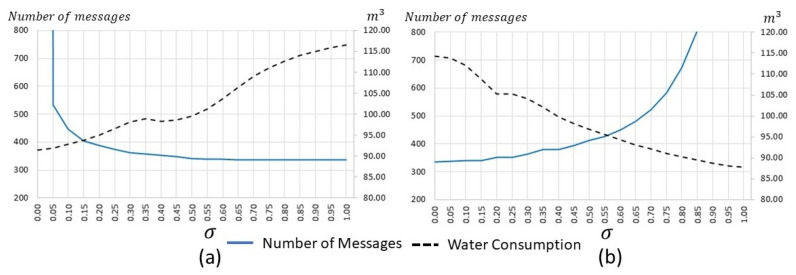
Estimated transmitted messages and water consumption: (**a**) Error-based triggering condition $\gamma_{1}$. (**b**) Lyapunov-based triggering condition $\gamma_{2}$.

**Figure 7 sensors-21-05541-f007:**
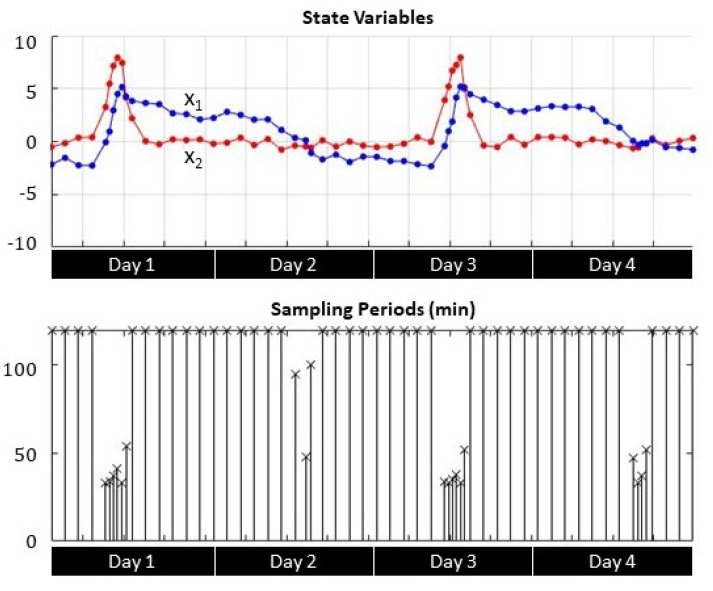
Self-triggered error-based state variable evolution and sampling periods.

**Figure 8 sensors-21-05541-f008:**
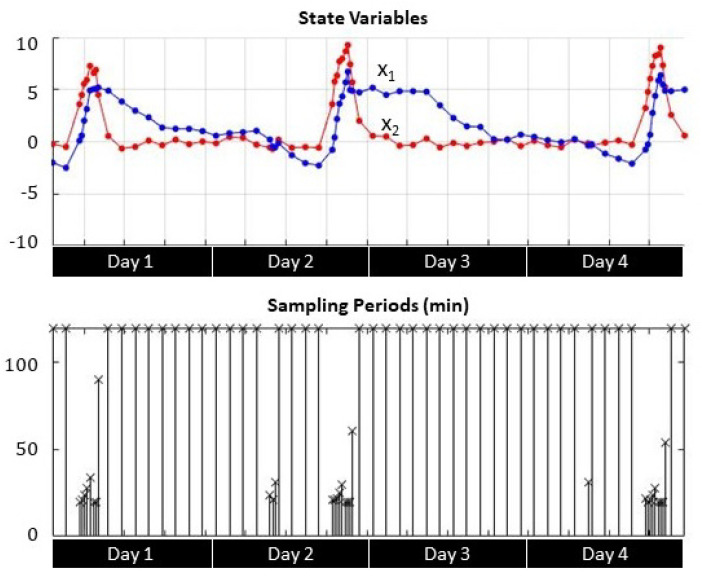
Self-triggered Lyapunov-based state variable evolution and sampling periods.

**Figure 9 sensors-21-05541-f009:**
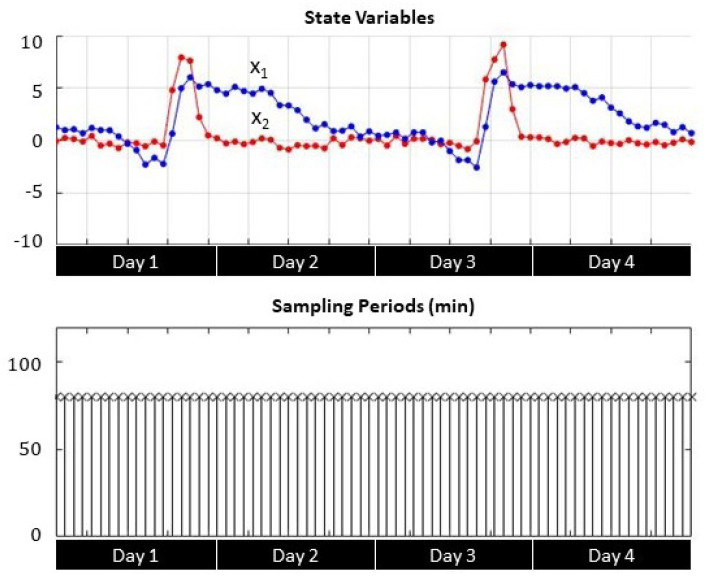
Periodical sampling (h = 80 min) state variable evolution and sampling periods.

**Figure 10 sensors-21-05541-f010:**
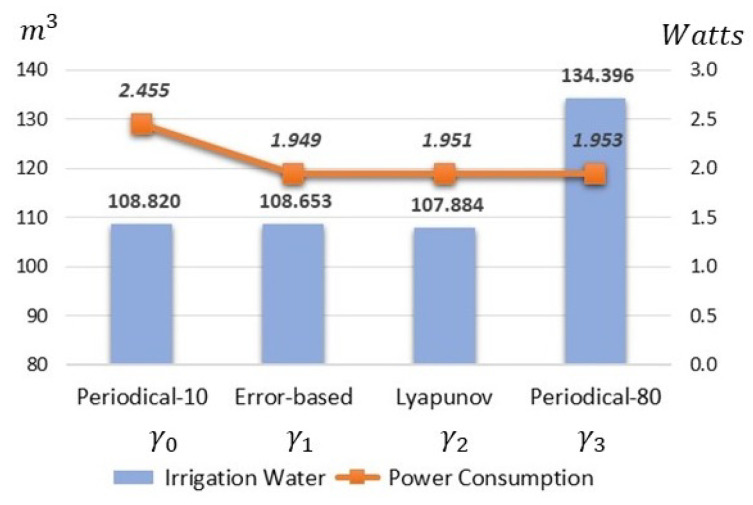
The estimated spent energy for the sensor nodes and water consumption for each irrigation area.

**Table 1 sensors-21-05541-t001:** Identified coefficient values for state–space model matrices according to the soil moisture region.

Matrix	Region	Coefficients		Matrix	Region	Coefficients	
Φ	Gravitational	0.98501	0.00001	Γ	Gravitational	0.00245	−0.00004
		0.00001	0.98491			0.00300	−0.00021
Φ	Available	0.99998	0.00003	Γ	Available	0.00136	−0.00062
		0.00001	0.98495			0.00324	−0.00049
Φ	Unavailable	0.99999	0.00002	Γ	Unavailable	0.00123	−0.00005
		0.00001	0.98488			0.00151	−0.00035

**Table 2 sensors-21-05541-t002:** Performance evaluation experimental results.

Sampling Strategy	Description	Communication Messages $J_{1}$	Water Consumption $J_{2}$	Irrigation Error $J_{3}$
$\gamma_{0}$	Periodical 10 min	4032	108.8200	7.4783
$\gamma_{1}$	Error-based	480	108.6529	7.3970
$\gamma_{2}$	Lyapunov-based	492	107.8835	7.4641
$\gamma_{3}$	Periodical 80 min	504	134.3956	8.4579

**Table 3 sensors-21-05541-t003:** The current measurements in the sensor node for each operation state.

Operation State	Current	Time
Data Acquisition	30 mA	450 ms
Data Transmission	360 mA	210 ms
Data Receiving	75 mA	210 ms
Data Processing	20 mA	0.1 ms
Standby Mode	0.57 mA	-

## Data Availability

Not applicable.

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
