# Peer review of "Energy-Efficient Wireless Communication Strategy for Precision Agriculture Irrigation Control"

_sensors, 2021, doi:10.3390/s21165541_

Round 1

Reviewer 1 Report

Strong aspects:

In this paper, the authors take care of the energy consumption in wireless sensor nodes by proposing energy efficient communication schema in the smart agriculture domain. More precisely, they propose an energy-efficient communication strategy for closed-loop control irrigation, implemented over a wireless sensor network, where soil moisture measurements are conducted by the sensing devices as event driven. Self-triggered algorithm estimates the occurrence of the next sampling period based on the process dynamics. They evaluated the proposed strategy in a pecan crop experimental field and compared it with periodical sampling implementations with the objective to reduce power consumption on network devices and still accomplish the system control objectives in terms of irrigation efficiency and water consumption.

The main contributions of this paper are the identification and validation of the process dynamics for the pecan crop irrigation system, the implementation of a model-based self-triggered control algorithm to decrease the number of communication messages while still accomplishing control objectives, and the estimation of the energy consumption improvement in the wireless sensor nodes due to the use of this control approach.

    • The challenges are well explained.
    • The paper is well written
    • The formulas as well as the structure of the approach are clear and easy to understand.
    • Good results despite the lack in the comparison part

Weak aspects:

Following are some remarks, recommendations and questions:

    • Half of the references are pre-2017. A lot of papers have been published in this topic in 2020-2021.
    • Use “event driven” instead of “aperiodically … when needed”
    • A comparison with one or more identified clear approaches from real references is needed (i.e compare the technique with other energy-efficient sampling rate adaptation technique and/or energy-efficient data reduction for transmission...)
    • It would be better to move the “evaluation parameters” subsection to the “Results and discussion” section.

Author Response

The attached document addresses point-by-point the comments, recommendations and questions provided by the reviewer 1.

Reviewer 2 Report

  1. The authors claim that efficient data communication and energy management strategies for the sensor devices have not been addressed explicitly. So I suggest the authors specify what scholars have done and have not done but still essential with relevant supporting reference regarding the data communication and energy management strategies for the sensor devices.

  1. Some presentations are so absolute and may not be proper. For example:

“To the best of our knowledge, no work so far has analyzed and evaluated the use of self-triggered control the practical in-site implementation of closed-loop irrigation systems within the precision agriculture domain.”

  1. The research gap is vague so it seems that the contributions are not salient. I suggest the authors summary the gaps based on the analysis of literature.

  1. In section 3.2, do you make some modifications about the existing algorithms?

  1. Some relevant reference about agricultural IoT may be beneficial for the authors. For example,

  • Verdouw C, Wolfert S, Tekinerdogan B. Internet of Things in agriculture[J]. CAB Reviews: Perspectives in Agriculture, Veterinary Science, Nutrition and Natural Resources, 2016, 11(35): 1-12.

  • Ruan J, Hu X, Huo X, et al. An IoT-based E-business model of intelligent vegetable greenhouses and its key operations management issues[J]. Neural Computing & Applications, 2020, 32(19).

  • Chen J, Yang A. Intelligent agriculture and its key technologies based on internet of things architecture[J]. IEEE Access, 2019, 7: 77134-77141.

  • Hu X, Sun L, Zhou Y, et al. Review of operational management in intelligent agriculture based on the Internet of Things[J]. Frontiers of Engineering Management, 2020, 7(3): 309-322.

Author Response

The attached document addresses point-by-point the comments, recommendations and questions provided by the reviewer 2.

Reviewer 3 Report

The paper proposes an energy-efficient communication strategy for✽
closed-loop control irrigation, implemented over a wireless sensor network, where soil moisture measurements are conducted by the sensing devices aperiodically and only when needed.
The paper presents new scientific elements and I recommend its publication.
In the attached file I made comments

Author Response

The attached document addresses point-by-point the comments, recommendations and questions provided by the reviewer 3.

Round 2

Reviewer 2 Report

My comments have been solved, so I recommend with publication.